# The Role of Serotoninomics in Neuropsychiatric Disorders: Anthranilic Acid in Schizophrenia

**DOI:** 10.3390/ijms26157124

**Published:** 2025-07-24

**Authors:** Katia L. Jiménez-García, José L. Cervantes-Escárcega, Gustavo Canul-Medina, Telma Lisboa-Nascimento, Francisco Jiménez-Trejo

**Affiliations:** 1Faculty of Medicine, National Autonomous University of Mexico, Circuito Escolar S/N, Ciudad Universitaria, Mexico City 04510, Mexico; katiajimenezz03@gmail.com; 2Angels of Pedregal Hospital, Mexico City 10700, Mexico; joscerv9@gmail.com; 3School of Medicine, Educational Centre Rodriguez Tamayo, Ticul 97862, Yucatan, Mexico; gustavoecm25@gmail.com; 4Bioengineering Postgraduate, University of Brazil, Itaquera, São Paulo 08230-030, Brazil; telmanascimento@ub.edu.br; 5National Institute of Paediatrics, Insurgentes Sur, Coyoacán, Mexico City 04530, Mexico

**Keywords:** serotoninomics, anthranilic acid, schizophrenia, serotonin, neuropsychiatric disorder

## Abstract

Serotoninomics is an expanding field that focuses on the comprehensive study of the serotoninergic system, including serotonin’s biosynthesis, metabolism, and regulation, as well as related scientific methodologies 5-hydroxytryptamine (5-HT). This field explores serotonin’s complex roles in various physiological and pathological contexts. The essential amino acid tryptophan (Trp) is a precursor for several metabolic and catabolic pathways, with the kynurenine (KYN) pathway being particularly significant, representing about 95% of Trp metabolism. In contrast, only a small portion (1–2%) of dietary Trp enters the serotonin pathway. Anthranilic acid (AA), a metabolite in the KYN pathway, has emerged as a potential biomarker and therapeutic target for schizophrenia. Elevated serum AA levels in patients with schizophrenia have been associated with neurotoxic effects and disruptions in neurotransmission, suggesting AA’s critical role in the disorder’s pathophysiology. Furthermore, the 5-HT_2A_ receptor’s involvement is particularly noteworthy, especially in relation to schizophrenia’s positive symptoms. Recent findings indicate that 5-HT_2A_ receptor hyperactivity is linked to positive symptoms of schizophrenia, such as hallucinations and delusions. This study investigates serotoninomics’ implications for neuropsychiatric disorders, focusing on AA in schizophrenia and analysing recent research on serotonin signalling pathways and AA’s neurochemical effects. Understanding the roles of the 5-HT_2A_ receptor and AA in neuropsychiatric disorders could lead to the development of more precise and less invasive diagnostic tools, specific therapeutic strategies, and improved clinical outcomes. Ongoing research is essential to uncover these pathways’ exact mechanisms and therapeutic potential, thereby advancing personalised medicine and innovative treatments in neuropsychiatry.

## 1. Introduction

In this era of omics sciences, our group aims to position serotoninomics as an emerging field focusing on the comprehensive study of the serotonergic system. This includes the biosynthesis, metabolism, transcription mechanisms, regulation, and function of serotonin or 5-hydroxytryptamine (5-HT; C_10_H_12_N_2_O) in various physiological and pathological contexts. It also encompasses the technical aspects and scientific methodologies for the study of the serotonin molecule [1,2]. The main methodologies that have been used or adapted as study tools for the field of serotonin in recent years include various advanced techniques such as mass spectrometry, high-performance liquid chromatography, and next-generation sequencing. Combining these strategies with emerging technologies, supported by advancements in laboratory instruments, will lead to a better understanding of the serotonergic system’s different roles across a broad range of biological systems, including neuropsychiatric disorders (see Table 1, taken from [2]).

Serotonin is a key neurotransmitter in the central nervous system that regulates multiple essential biological processes, such as mood, the sleep–wake cycle, appetite, memory, and cardiovascular homeostasis. Alterations in this molecule are known to be associated with neuropsychiatric disorders such as depression and anxiety, as well as metabolic and cardiovascular diseases. This physiological prominence arises not only from serotonin’s presence in multiple systems but also from its interaction with a diverse family of serotonergic receptors, such as 5-HT_1A_ and 5-HT_2C_, whose proper activity is fundamental in triggering specific responses through second messenger signalling cascades, such as cyclic AMP or calcium. These mechanisms are essential in the modulation of neuronal circuits associated with complex physiological processes [2,3].

Moreover, serotonin is synthesised from L-tryptophan (see below), an essential amino acid whose availability directly influences serotonin production and the generation of metabolites such as kynurenic acid, which is implicated in neuroinflammatory processes. This interconnection between metabolites and signalling pathways highlights the serotonergic system’s relevance in regulating critical biological functions and in the genesis of pathologies when its balance is disrupted [3].

As a result of this growing relevance, we coined the term “serotoninomics” in 2015 [1], and, since then, there has been growing interest in exploring its implications in various fields of biological sciences, including neuroscience and neuropsychiatric disorders. As a whole, this will contribute to finding precise answers regarding basic, clinical, and translational research related to serotonin, just as the emerging medical and “omics” sciences have already achieved [4].

Additionally, it involves adapting and positioning new terms and concepts that arise in the world of serotonin with the help of the International Society for Serotonin Research (ISSR; formerly Serotonin Club) to standardise and disseminate this concept related to this indolamine [2]. The analysis and study of serotonin’s actions at different levels of the structure and function of cells, tissues, or even complete organisms will allow us to achieve a better understanding of serotonin’s role in human health and disease [4,5].

Furthermore, stronger efforts must be made to achieve molecular and pharmacological treatments that are long-lasting and lack secondary effects if we wish to find a cure for neuropsychiatric disorders, such as anxiety and depression, in which the serotonergic system is implicated. Understanding how different drugs (selective serotonin reuptake inhibitors (SSRIs), agonists, antagonists, and psychedelics) participate in mental illness and addiction processes is crucial in restoring such patients’ mental homeostasis. Animal models, genetically encoded serotonergic sensors, and non-invasive imaging techniques are promising approaches that will gain importance in the years to come [6,7].

As we have previously described, serotonin’s involvement in biological processes is extensive, and it is evident that much research remains to be conducted to further expand its physiological significance and achieve a more complete understanding of its role in pathologies where its balance has been altered, such as psychiatric disorders [3,8]. Training on recent technological innovations with the help of artificial intelligence (AI) and new applications will allow us to gain new perspectives on brain function in disease, which in turn will enable us to perform better therapeutic interventions [2,7].

Together, this new holistic approach will allow us to gain a deeper understanding of how serotonin and its components influence various physiological and brain functions and, in this case, their implications for neuropsychiatric disorders such as schizophrenia. The aim is to address neuropsychiatric disorders by developing faster and more accurate diagnostics, managing targeted therapies with better drugs without side effects, and achieving more precise clinical outcomes for all those patients who require immediate and long-term care [6,7].

Schizophrenia is a complex neuropsychiatric disorder that affects millions of people worldwide (see below). Various studies have shown that serotonergic dysfunction plays a crucial role in this disorder’s pathophysiology. Identifying and studying key metabolites, such as anthranilic acid (AA), can provide new insights into the underlying molecular mechanisms and open the door to more effective treatments [9].

In the 2020s, it is important to position the concept of serotoninomics and expand it to encompass new perspectives and applications [2]. Recent studies have indicated that AA, a metabolite of the amino acid tryptophan (L-Trp), could play a significant role in modulating the altered serotonergic pathways in schizophrenia (as detailed below). This approach not only opens up new perspectives for basic research but also raises the possibility of developing specific biomarkers for the early diagnosis and monitoring of the disorder [9,10,11].

Integrating serotoninomics into the study of neuropsychiatric disorders could revolutionise our understanding of and therapeutic approach to disorders such as schizophrenia. It will also help contribute to other omics areas, allowing us to increase our understanding of biological processes from the molecular level to complex organisms and their brain functions, as well as the resolution of psychiatric disorders. By expanding our knowledge of serotonin, we can move one step closer to improving patients’ quality of life and providing more effective and personalised medical solutions.

## 2. Psychiatric Disorders

Mental health disorders, such as schizophrenia in its various forms, are severe illnesses that affect patients’ functioning in all areas of life. Common symptoms include hallucinations, delusions, and cognitive impairments. Severe psychiatric disorders have a devastating impact on psychosocial functioning and relationships within the family, social, and work environments. Schizophrenia is a complex, multifaceted, and multi-aetiological early-onset debilitating disorder that affects approximately 24 million people worldwide, or between 0.34% and 0.44% of the population [12]. Moreover, individuals with schizophrenic parents or siblings have a higher risk of developing the disorder due to its hereditary component (between 8% and 12%) [6,7]. Some studies conceptualise schizophrenia as a disorder of functional “dysconnectivity” [6] or a “synapse disorder” [9], affecting the serotonin quantum release machinery within the synapse during its synthesis, its interaction with its receptors, or its correlation with other neurotransmitters (i.e., dopamine, glutamate, and gamma-aminobutyric acid (GABA)) [13,14].

Dopamine has long been considered the principal neurotransmitter involved in schizophrenia’s pathophysiology, playing a critical role in many of the symptoms expressed by patients with this disorder. The dopamine hypothesis posits that the hyperactivity of dopamine transmission in certain brain regions, such as the mesolimbic pathway, contributes to positive symptoms like hallucinations and delusions, while hypoactivity in the prefrontal cortex is associated with negative and cognitive symptoms, such as anhedonia and impaired executive function [14,15]. This dysregulation in the dopamine pathways is believed to be a core component of the disorder, influencing various aspects of its presentation and progression [16,17]. Antipsychotic medications, which primarily target dopamine receptors, have been the mainstay of treatment for schizophrenia, aiming to alleviate positive symptoms by reducing dopaminergic activity. However, these medications often come with significant side effects and may not fully address the cognitive and negative symptoms that patients experience [15,17]. Understanding the complex interplay between dopamine and other neurotransmitter systems, including serotonin, is crucial in developing more effective and comprehensive schizophrenia treatment strategies.

The serotonergic system modulates the release of dopamine in key areas of the brain, such as the prefrontal cortex and the nucleus accumbens, regions involved in schizophrenia’s neurochemical alterations. This interaction highlights the importance of balanced serotonergic activity in mitigating dopaminergic hyperactivity, which is characteristic of positive symptoms, and hypodopaminergic states, associated with negative and cognitive symptoms [18].

Although its aetiology is not fully understood, in certain forms of schizophrenia, the 5-HT_2A_ receptor may be involved in alterations in serotonin signalling pathways and effects, playing a crucial role in the disorder’s pathogenesis, particularly in the presentation of positive symptoms. Meanwhile, 5-HT_1A_, which has been widely studied for its role in schizophrenia, is related to the regulation of anxiety and emotions, areas affected in patients with schizophrenia [19,20].

Therefore, serotoninomics could offer important information to address psychiatric disorders such as schizophrenia. Moreover, physicians and the pharmaceutical industry still lack definitive curative treatments, and patients and their relatives or caregivers face repeated failures in daily life [9]. Currently, few medications significantly improve the health status of patients with schizophrenia. Some antipsychotic medications can help to control symptoms such as hallucinations and delusions, but they do not always fully address the cognitive and emotional problems that patients experience. Additionally, each person may respond differently to treatments, making it challenging to find an effective solution.

## 3. The Role of the 5-HT_2A_ Receptor and Other Serotonergic Pathways in Psychiatric Disorders

The Kossatz group has significantly contributed to elucidating 5-HT_2A_ receptor-mediated pathways’ roles in paranoid schizophrenia-like behavioural responses through a multidisciplinary approach [21]. Their methodology has integrated computational models, in vitro and in vivo experiments, and postmortem human brain studies. In the latter studies, their postmortem investigations showed the increased functional activity of the 5-HT_2A_ receptor in the brains of patients with schizophrenia. It is known that 5-HT_2A_ receptor hyperactivity is associated with positive symptoms, such as hallucinations and delusions, making this receptor a critical and prominent target for the treatment of this type of schizophrenia.

The distinct signalling pathways mediated by the 5-HT_2A_ receptor, such as those involving Gaq and Gi/o proteins, not only regulate memory deficits and psychosis-related behaviours but also influence other core aspects of schizophrenia, including sensory processing and behavioural flexibility [21]. Hyperactivity of the 5-HT_2A_ receptor, particularly in the Gi/o protein pathway, is a hallmark of prohallucinogenic potential, underscoring the receptor’s pivotal role in schizophrenia. Antagonists targeting this receptor, such as atypical antipsychotics, attenuate its hyperactivity, offering relief from positive symptoms like hallucinations and delusions.

Additionally, in paranoid-like schizophrenia, the 5-HT_2A_ receptor exhibits selective functional hyperactivity in the signalling pathway involving Gi/o proteins. This hyperactivity is considered a hallmark of prohallucinogenic potential [21,22]. However, more studies on this type of receptor are needed to address this important mental disorder more comprehensively and rapidly. Researchers must seek to identify new compounds related to the serotonergic system that may be more effective and have fewer side effects than conventional antipsychotics. Current research must be directed towards finding new biomarkers related to the 5-HT_2A_ receptor to identify at-risk patients and develop early treatments before the end of this decade. These findings underscore not only the importance of the 5-HT_2A_ receptor but also its potential to serve as a model in exploring the broader serotonergic system’s contributions to psychiatric disorders. Altogether, these data highlight the receptor’s importance in schizophrenia’s pathology and the need to continue researching to improve the available treatments.

In addition to the 5-HT_2A_ receptor’s critical role in schizophrenia’s pathology, other serotonergic receptors and enzymes also contribute significantly to the understanding of psychiatric disorders. The 5-HT_1A_ receptor, for instance, is known to play a role in anxiety and depression. Alterations in 5-HT_1A_ receptor expression and function have been linked to these conditions, suggesting that targeting this receptor could provide therapeutic benefits. Additionally, the 5-HT_1A_ receptor has been associated with cognitive functions, and its modulation may influence memory and learning processes, which are often impaired in psychiatric disorders [23].

Enzymes involved in serotonin synthesis, such as tryptophan hydroxylase 2 (TPH2, the central isoform), are also critical in understanding psychiatric disorders. TPH2 is the rate-limiting enzyme in the synthesis of serotonin in the brain, and genetic variations in the TPH2 gene have been associated with altered serotonin production and an increased risk of psychiatric disorders such as major depressive disorder and bipolar disorder [24]. Studies have shown that polymorphisms in the TPH2 gene can lead to reduced enzyme activity, resulting in lower serotonin levels and contributing to the pathogenesis of these disorders.

Moreover, the interaction between serotonin and other neurotransmitter systems, such as dopamine and glutamate, is crucial in the context of psychiatric disorders. The dopaminergic system, as previously discussed, is heavily implicated in schizophrenia. However, the interplay between serotonin and dopamine can influence various aspects of the disorder, including symptom severity and the response to treatment. Similarly, the glutamatergic system, which involves the neurotransmitter glutamate, has been linked to cognitive deficits and negative symptoms in schizophrenia. Modulating serotonergic pathways may have downstream effects on glutamate signalling, providing a potential avenue for therapeutic intervention [13,15].

In summary, the study of serotonergic receptors, enzymes, and their interactions with other neurotransmitter systems is essential in advancing our understanding of psychiatric disorders. By exploring these pathways in greater detail, researchers can develop more effective and targeted treatments, ultimately improving the quality of life of patients suffering from these debilitating conditions.

## 4. The Role of Anthranilic Acid in Schizophrenia

A key component of our future serotoninomics research is exploring the range of potential implications of AA. This metabolite plays a fundamental role in the kynurenine pathway, which is closely related to serotonin metabolism. Research has suggested that an imbalance in AA levels may contribute to the neurochemical abnormalities observed in patients with schizophrenia [11,25]. This imbalance may arise from dysregulation in the kynurenine pathway—specifically, the overactivation of enzymes such as kynureninase, which increases the production of AA, or the disruption of downstream metabolites such as 3-hydroxyanthranilic acid (3-HAA). Therefore, AA has recently attracted attention as both a potential biomarker and a therapeutic target for schizophrenia.

L-tryptophan (L-Trp), as an essential amino acid, enters the human body through the diet. It serves as a precursor for both serotonin and melatonin and is implicated in human neuropsychiatric conditions, including schizophrenia [26]. The metabolism of L-Trp also contributes to the kynurenine (Kyn) pathway, which produces several important metabolites for the brain and the immune system, including AA [25].

Recently, the dysregulation of the tryptophan–kynurenine pathway, particularly the overproduction of quinolinic acid (KYNA), has been implicated in the pathogenesis of schizophrenia and depression. However, AA’s role as another secondary metabolite of kynurenine remains less explored [27,28]. Serotonin synthesis may be altered either by the disruption of the enzyme TPH2 and its associated pathways or by the diversion of L-Trp into the kynurenine pathway. In this pathway, L-Trp is converted into N-formylkynurenine by the enzymes tryptophan 2,3-dioxygenase (TDO) and indoleamine 2,3-dioxygenase (IDO) [27,29]. N-formylkynurenine is then rapidly transformed into kynurenine by the enzyme formamidase.

Additionally, one pathway of kynurenine metabolism produces AA via kynureninase. Overactive kynureninase could lead to the excessive accumulation of AA, disrupting the balance of metabolites within the kynurenine pathway. This dysregulation has been linked to oxidative stress and neuroinflammation, processes implicated in schizophrenia’s pathophysiology [28]. Recent studies have shown a significant increase in serum AA concentrations in patients with schizophrenia compared to control subjects. This elevation in AA could be related to the downregulation of mitochondrial enzymes and the increased formation of 3-hydroxyanthranilic acid (3-HAA), a potent generator of free radicals and glutamatergic agonists. Its accumulation in key brain nuclei that regulate behaviour can lead to neurodegenerative diseases. This accumulation is particularly problematic because it can disrupt normal neurotransmission and lead to the deterioration of cognitive and behavioural functions [28,29].

Furthermore, new schizophrenia treatments are urgently needed. AA, known to be a G-protein-coupled receptor 109A (GPR109A) agonist, could activate this receptor and offer beneficial effects, such as the preservation of myelin integrity and improvements in cognitive function. This is due to its ability to inhibit cytosolic phospholipase A2 (cPLA2), an enzyme implicated in cognitive impairment associated with schizophrenia, as it breaks down myelin. This enzyme is upregulated in individuals with schizophrenia and in people at high risk of developing psychosis. However, further studies are required to elucidate these potential benefits, given the current antipsychotic medications’ limited clinical efficacy. The expression of GPR109A could represent a new endophenotype of schizophrenia particularly associated with cognitive impairment, although this requires more exhaustive evaluation [30].

The AA and kynurenine pathways are essential for tryptophan degradation and the production of metabolites that modulate mental health and the immune system. These findings suggest that AA could serve as a biological marker, or endophenotype, for a subgroup of patients with schizophrenia and offer new avenues for therapeutic intervention [11]. Understanding the exact mechanisms by which AA contributes to schizophrenia’s pathophysiology will be crucial in developing targeted treatments. By identifying and modulating specific pathways involving AA and related metabolites, it may be possible to reduce symptoms and improve the quality of life of patients with this type of schizophrenia [28,30].

AA’s effects on the immune system and its potential to influence neurogenesis highlight its importance in the broader context of schizophrenia research. Exploring its interactions with other neurotransmitters and identifying new biomarkers could provide a more comprehensive understanding and lead to more effective therapeutic strategies. Recent clinical and experimental studies have begun to uncover these complex roles, underscoring the need for continued investigation [10,28].

Furthermore, AA’s interaction with other neurotransmitter systems, such as glutamate, is also worth exploring. Glutamate, the primary excitatory neurotransmitter in the brain, plays a crucial role in synaptic plasticity and cognitive functions. Dysregulation of the glutamatergic system has been implicated in the cognitive deficits observed in schizophrenia [13]. By examining the interplay between AA and glutamate, researchers can gain insights into potential therapeutic targets to address both the cognitive and psychotic symptoms of schizophrenia.

Finally, the kynurenine pathway’s role in modulating immune responses is another critical area of investigation. Inflammatory processes have been associated with schizophrenia’s pathophysiology, and AA’s influence on these processes could provide new avenues for therapeutic intervention. By understanding how AA modulates inflammation and immune responses, it may be possible to develop treatments that not only target the neurochemical abnormalities but also address the underlying immune dysregulation in schizophrenia [10,11,25]. Understanding these mechanisms is essential to determine how an AA imbalance contributes to both neurochemical disruptions and the clinical presentation of schizophrenia, thereby guiding future research into targeted interventions.

## 5. Perspectives

The field of serotoninomics requires a more comprehensive approach from the international community to provide valuable information on the complex interaction between serotonin and neuropsychiatric disorders, particularly schizophrenia. Understanding the integral role of serotonin, its receptors, and related metabolites is crucial in quickly developing specific and effective treatments for these conditions. The significant findings on the 5-HT_2A_ receptor in schizophrenia’s pathophysiology, particularly related to positive symptoms, underline the importance of multidisciplinary approaches in elucidating these mechanisms and paving the way for more refined therapeutic strategies.

Furthermore, the study of anthranilic acid as part of the kynurenine pathway reveals its potential as a biomarker and therapeutic target for schizophrenia. The imbalance in AA levels and its impact on neurochemical processes underscore the need for further research on its role and the development of specific interventions to modulate this pathway. Elevated AA concentrations have been associated with neurotoxic effects and disruptions in normal neurotransmission, highlighting the importance of maintaining AA homeostasis for cognitive and behavioural health. AA’s role as a GPR109A agonist also suggests that it could represent a new endophenotype of schizophrenia, particularly associated with cognitive impairment, requiring more exhaustive evaluation.

Ongoing research on serotonin metabolism and its broader implications for mental health demonstrates that a deeper understanding of these biochemical pathways can lead to more effective treatments and improved quality of life for patients. Future studies focusing on the modulation of serotonin receptors and related metabolites, such as AA, will be essential in advancing the therapeutic options and achieving better clinical outcomes for individuals with schizophrenia.

Additionally, the integration of advanced technologies such as AI and machine learning into serotoninomics research holds promise in accelerating discoveries and enhancing the precision of therapeutic approaches. These technologies can facilitate the identification of novel biomarkers, predict treatment responses, and optimise drug development processes, ultimately contributing to the advancement of personalised medicine in psychiatry.

## 6. Conclusions

This comprehensive review highlights how anthranilic acid and serotoninomics can play essential roles in the treatment of schizophrenia, as well as representing a new endophenotype of the disorder, opening up new avenues for research and the development of innovative therapies. By continuing to unravel the complexities of serotonin and its related pathways, researchers and clinicians may develop more precise and effective interventions, ultimately contributing to the alleviation of symptoms and improvements in the lives of those affected by schizophrenia. The international scientific community’s collaborative efforts, combined with the application of cutting-edge technologies, will be pivotal in achieving these goals and advancing our understanding of neuropsychiatric disorders.

## Figures and Tables

**Table 1 ijms-26-07124-t001:** Methodologies used or adapted as study tools in the field of serotonin in the past and present.

(1) Histochemistry for indolamines or Falck–Hillarp method
(2) Brightfield immunocytochemistry and immunohistochemistry or direct or indirect immunofluorescence, single or multiplex
(3) Brightfield and fluorescence in situ hybridisation
(4) Super-resolution microscopy
(5) Electron microscopy and cryo-electron microscopy
(6) Fluorescence spectroscopy
(7) Flow cytometry
(8) Polymerase chain reaction (PCR) and reverse transcription polymerase chain reaction (RT-PCR)
(9) Molecular genotyping
(10) Enzyme-linked immunosorbent assay (ELISA)
(11) Optogenetics
(12) Transgenic models
(13) Behavioural trials (using transgenic models, agonists and/or antagonists, or other molecules acting over serotonergic pathway elements)
(14) Cell culture and three-dimensional (3D) printed models
(15) Chromatography
(16) Western blotting
(17) Two-dimensional (2D) gel electrophoresis
(18) Clustered regularly interspaced short palindromic repeats (CRISPR) gene editing
(19) Sanger or high-performance sequencing
(20) Single-cell transcriptomic profiling
(21) Bulk-tissue ribonucleic acid (RNA) sequencing and single-cell RNA sequencing
(22) Electrophysiology
(23) X-ray (radiograph) crystallography
(24) Mass spectrometry
(25) Drug delivery via nanoparticles
(26) Genomic analysis
(27) Meta-analysis
(28) Bioinformatics

This table provides an overview of the methodologies employed in serotonin research over the years. It encompasses a variety of tools and techniques used to investigate serotonin’s role in various neuropsychiatric disorders, including schizophrenia. The table highlights these methodologies’ evolution from traditional approaches to cutting-edge technologies, illustrating the progress made in the field of serotoninomics. Each listed method has contributed to our current understanding of serotonin’s complex interactions and their implications for mental health (taken from [2]).

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
