# Peer review of "The Role of Serotoninomics in Neuropsychiatric Disorders: Anthranilic Acid in Schizophrenia"

_ijms, 2025, doi:10.3390/ijms26157124_

Round 1

Reviewer 1 Report

Comments and Suggestions for Authors

Please, delete expressions such as “male reproduction” and “reproductive”. There is no point to cite those topics in this article.

Please, avoid using the concept of psychiatric diseases. Use instead psychiatric disorders.

Please, avoid using the concept of mental illnesses. Use instead psychiatric disorders.

Please, rephrase “playing a crucial role in the pathogenesis of the disease, particularly of the paranoid type.” Do not use “disease” and do not use “paranoid type”. Use “disorder instead” and “positive symptoms” instead. Please delete “paranoid type” and “paranoid subtype”. The concepts are inconsistent. And they were abandoned in recent nosological classifications (e.g. ICD11).

The list of abbreviations is incomplete.

Please, explain all acronyms, such as PCR, RT-PCR, 3D, 2D, CRISPR, RNA, X-ray, SSRI, i.e., GABA, AI, etc.

Author Response

Comments and Suggestions for Authors R1

Dear Reviewer 1,

We sincerely thank you for your detailed comments and valuable suggestions to improve the manuscript. Your observations have significantly helped us refine our work, and we have carefully addressed each point raised. Below is our response to your comments:

Please, delete expressions such as “male reproduction” and “reproductive”. There is no point to cite those topics in this article.

  1. Response: These expressions have been removed from the manuscript as suggested.

Please, avoid using the concept of psychiatric diseases. Use instead psychiatric disorders.

  1. Response: All occurrences of “psychiatric diseases” have been replaced with “psychiatric disorders” in the revised manuscript.

Please, avoid using the concept of mental illnesses. Use instead psychiatric disorders.

  1. Response: The term “mental illnesses” has been replaced with “psychiatric disorders” throughout the manuscript.

Please, rephrase “playing a crucial role in the pathogenesis of the disease, particularly of the paranoid type.” Do not use “disease” and do not use “paranoid type”. Use “disorder instead” and “positive symptoms” instead. Please delete “paranoid type” and “paranoid subtype”. The concepts are inconsistent. And they were abandoned in recent nosological classifications (e.g. ICD11).

  1. Response: The sentence has been rephrased. Additionally, all references to “paranoid type” and “paranoid subtype” have been removed to align with the updated nosological classifications, such as ICD-11.

The list of abbreviations is incomplete.

  1. Response: The list of abbreviations has been reviewed and completed. Any missing acronyms have been added to the list.

Please, explain all acronyms, such as PCR, RT-PCR, 3D, 2D, CRISPR, RNA, X-ray, SSRI, i.e., GABA, AI, etc.

  1. Response: All acronyms, including those mentioned (PCR, RT-PCR, 3D, 2D, CRISPR, RNA, X-ray, SSRI, GABA, AI), have been fully explained in the revised manuscript.

Reviewer 2 Report

Comments and Suggestions for Authors

This review manuscript by Jimernez-Garcia focuses on the serotoninomics’ implication in schizophrenia. They first framed some works related to serotoninomics and then linked the serotonin 5-HT2a receptor to psychiatric disorder, mainly schizophrenia. Finally, they discussed some studies connecting one of the serotonin metabolisms anthranilic acid with schizophrenia. While the topic of the review is potentially interesting, the current manuscript provides limited and fewer descriptions of the main question. Here are my detailed comments:

The introduction provided limited background on why serotonin and its components are essential. The authors mainly reiterate the importance of serotonin but do not provide a  detailed and deeper discussion. 

The second part of the manuscript focuses on one of psychiatric disorders, schizophrenia and briefly touches on dopamine and then serotonin. However, without detailed discussion about how the serotonin system contributes to the process, it's hard for the readers to understand these connections.

This also applies to the section about the 5-HT2a receptor. While the authors state the importance of 5-HT2a, insufficient information was provided by the authors, making it difficult to see those connections.

Lastly, the authors said a potential future direction is to study the anthranilic acid. However, this could not be supported by the information provided. Like, how the imbalance of AA is introduced? How does this contribute to schizophrenia? There are also lots of things the authors mentioned, but mostly of them were just briefly mentioned without depth discussion.

Author Response

Comments and Suggestions for Authors R2

Dear Reviewer 2,

We sincerely thank you for your insightful comments and constructive suggestions regarding the manuscript. We have carefully reviewed each of your points and revised the manuscript to address your observations. Below are our responses to your detailed comments:

The introduction provided limited background on why serotonin and its components are essential. The authors mainly reiterate the importance of serotonin but do not provide a detailed and deeper discussion. 

  1. Response: The introduction has been expanded to provide a deeper and more detailed background on the importance of serotonin and its components.

The second part of the manuscript focuses on one of psychiatric disorders, schizophrenia and briefly touches on dopamine and then serotonin. However, without detailed discussion about how the serotonin system contributes to the process, it's hard for the readers to understand these connections.

  1. Response: The second part of the manuscript has been revised to include a more thorough discussion of the serotonin system and its contribution to the process of schizophrenia, to better clarify the connections mentioned.

This also applies to the section about the 5-HT2a receptor. While the authors state the importance of 5-HT2a, insufficient information was provided by the authors, making it difficult to see those connections.

  1. Response: The section on the 5-HT2A receptor has been expanded with additional information to clearly illustrate its importance and the connections to psychiatric disorders.

Lastly, the authors said a potential future direction is to study the anthranilic acid. However, this could not be supported by the information provided. Like, how the imbalance of AA is introduced? How does this contribute to schizophrenia? There are also lots of things the authors mentioned, but mostly of them were just briefly mentioned without depth discussion.

  1. Response: The discussion on anthranilic acid has been elaborated to include how its imbalance is introduced and its potential contribution to schizophrenia, along with more depth and supporting details. Additionally, we have introduced references to support the new content in this section, ensuring greater clarity and alignment with the current literature.

We are confident that these changes address your concerns and enhance the manuscript's clarity and depth. Once again, we sincerely appreciate your valuable insights.